# Feeling tired versus feeling relaxed: Two faces of low physiological arousal

**Sarah Steghaus** [ID] *, **Christian H. Poth** [ID]

Neuro-cognitive Psychology and Center of Cognitive Interaction Technology (CITEC), Bielefeld University, Bielefeld, Germany

* sarah.steghaus@uni-bielefeld.de

## Abstract

Human well-being and functioning depend on two fundamental mental states: Relaxation and sleepiness. Relaxation and sleepiness are both assumed to be states of low physiological arousal and negatively correlated. However, it is still unclear how consistent this negative relationship is across different settings and whether it changes before and after an intervention. Here we investigated this intricate relationship between subjective momentary sleepiness and relaxation states by meta-analytically analyzing several data sets from studies using the Relaxation State Questionnaire. We discovered that subjective sleepiness and relaxation were in fact anti-correlated pre-intervention. This anti-correlation provides a quantitative dissociation between sleepiness and relaxation. Thus, even though sleepiness and relaxation both implicate a low arousal level, the two mental states are subjectively experienced in a qualitatively different fashion, and thus reflect distinct underlying constructs. For the post-intervention relationship, this negative correlation could not be consistently found. This indicates that there are aspects of the experimental setting or intervention that introduce changes in the dynamics of the relationship of the two constructs.

**Data Availability Statement:** All data and the data analysis code are made available on the OSF (https://osf.io/34bsa/?view_only=391e8b016365446ba8b07b316cbd1067).

## Introduction

In our fast-paced world, the importance of relaxation has gained attention, as stress and stress related problems are on the rise. According to an APA-report, 76% of adults have experienced "health impacts due to stress in the prior month" [1]. With stress becoming a prevalent concern impacting mental and physical well-being, the significance of relaxation and relaxation practices to counteract these problems, cannot be overstated. Even though there have been numerous studies on the positive effect of relaxation and relaxation exercises [e.g., 2–4], it is difficult to find a definition of the construct *relaxation*. Relaxation has many variations and contains physical, neurological and psychological components [5, chapter 27]. Usually, relaxation implies a reduction of arousal, somewhat opposed to the stress related fight-or-flight response [6, 7, chapter 3]. Understanding relaxation in the current context not only addresses the immediate challenges of stress but also aligns with broader efforts to enhance overall quality of life in a rapidly evolving society [8].

One equally important and related construct is sleepiness. Similar to relaxation, sleepiness encompasses different facets, but overall it can be defined as an increased propensity to fall

**Funding:** The author(s) received no specific funding for this work.

**Competing interests:** The authors have declared that no competing interests exist.

asleep [9, 10]. Sleepiness is also correlated with both stress and psychological well-being [11, 12]. Experiencing stress in everyday life not only directly impacts individuals' health, but also results in poorer sleep quality and increased daytime sleepiness [13]. Daytime sleepiness directly impacts well-being and cognitive performance and may also lead to accidents and other problems [9, 14]. Thus, delving into the multifaceted implications of both relaxation and sleepiness may be crucial for fostering individual well-being and resilience in the face of modern stressors.

Intriguingly, sleepiness is, like relaxation, connected to a reduced arousal level. Arousal is referred to as a "generalized bodily activation" [15]. It could already be demonstrated in several experiments and settings that high states of arousal (e.g., externally elicited by alerting signals) can influence perception, cognition, and action [e.g., 16–18]. Low arousal levels are indicated by less activation; therefore, it is quite intuitive that individuals in a relaxed state, but also in a sleepy state, have low arousal levels. However, how these low arousal levels influence cognition has not yet received as much attention as the high arousal mechanisms.

## Sleepiness and relaxation

The basis for understanding the intriguing inverse relationship between sleepiness and relaxation can be traced back to works in mood and arousal research. Early works in this field proposed that relaxation and sleepiness are on the same spectrum of low arousal [for an overview see 19, chapter 3]. Sleepiness or tiredness was considered an extreme form of low arousal or low activation [20, 21], even lower than relaxation. However, one can easily imagine cases where this is not true. People often experience extreme tiredness without being relaxed at all, e.g., exhaustion. Also, it is perfectly plausible to be relaxed and not sleepy but rather mindful, calm, and concentrated. In line with this subjective experience, are works that propose distinct aspects of low arousal and emphasize the multifaceted nature of it. Thayer (1989) proposed two distinct systems of arousal: energetic arousal (EA) and tense arousal (TA). EA describes the subjective experience of wakefulness and alertness, with a decline leading to tiredness and diminished motivation for action. In contrast, TA is associated with feelings of tension, anxiety, stress, and fear. Interestingly, this system can also lead to mobilized action despite the presence of negative emotions. A decline in TA, however, is aligned with relaxation and a reduction in the physiological markers of stress. Furthermore, the two dimensions can interact in different ways to produce various combinations of mental states [19]. EA and TA may be categorized into high (+) and low (-) states, allowing four different combinations of EA and TA. Table 1 shows a fourfold table with all possible combinations, giving everyday situations where they may occur.

**Table 1. Fourfold table with arousal combinations.**

|  |  | TA+ | TA- |
|---|---|---|---|
| EA+ |  | *Challenge or high-stakes scenario* | *Enjoyment or pleasure* |
|  |  | Individuals are very engaged, involved and energetic, but also under a certain degree of pressure, urgency or stress | Individuals are energetic and positive, but relaxed and calm at the same time |
|  |  | Examples: Public speaking, important exam, competitive sports | Examples: Non-competitive sports, playing games, pursuing a hobby |
| EA- |  | *Anxiousness* | *Passivity* |
|  |  | Individuals are tense, nervous and/ or worrying, but at the same time without energy | Individuals are experiencing a lack of energy and enthusiasm while also experience no tension |
|  |  | Examples: Threatening or overwhelming situations | Examples: Lack of motivation, apathy, indifference |

TA+ = high tense arousal. TA- = low tense arousal. EA+ = high energetic arousal. EA- = low energetic arousal.

Both systems of arousal—energetic and tense—are rooted in intricate biochemical processes within the body. The intricate interplay of neurotransmitter systems, particularly the adrenergic pathway, has been implicated in governing both energetic and tense arousal. The autonomic nervous system, with its sympathetic and parasympathetic branches, also plays an essential role in shaping the physiological manifestations of these states [19].

A study by Huelsman et al. (1998) served as an early precursor, highlighting the negative correlation ($r = -.46$) between sleepiness and relaxation. Interestingly, the study considers relaxation and sleepiness to be traits and, while a factor analysis marked them as distinct factors, textual they are labeled both as "low affectivity". Later work by Schimmack and Grob [22] pointed out that the two forms of arousal (energetic and tense arousal) are indeed two *independent* factors and not just facets of one main arousal dimension.

This negative correlation between sleepiness and relaxation was also found in the construction of the Relaxation State Questionnaire [RSQ; 23]. The RSQ is a novel tool assessing subjective relaxation and its short-term changes. Measuring short-term effects and changes in relaxation has long been neglected, since the focus of the research was mainly on long-term relaxation and trait-like conceptions rather than situation-dependent state. However, to be able to account for changes introduced by e.g., experimental interventions, a measurement sensitive to short-term changes is essential. The RSQ solves this issue and is able to differentiate relaxation into 3 subscales and one sleepiness scale. All scales have been established by two independent factor analyses. Relaxation aspects covered by the RSQ are divided into the categories: Muscle Relaxation (focusing on changes e.g., evoked by Progressive Muscle Relaxation exercises [24, 25]), Cardiovascular Relaxation (capturing individually observable body-related changes in breathing and heart rate) and General Relaxation (meaning overall relaxation items with a high face-validity, such as answering whether or not one feels relaxed) [23]. The two independent factorial analyses also revealed a negative correlation between sleepiness and the three highly correlated factors of relaxation (Muscle Relaxation, Cardiovascular Relaxation, and General Relaxation). Therefore, the RSQ provides an economic and easily applicable tool to measure both relaxation and sleepiness at the same time.

## Research questions and objectives

Further understanding the complex relationship between sleepiness and the facets of relaxation can impact both the theoretical understanding of the constructs and have a variety of practical implications, e.g., concerning stress management or well-being. While state questionnaires (and other measures) are frequently used in research, especially in experimental settings, the underlying concepts are often not clearly distinguished. This may lead to confusion between similar-sounding terms with different meanings (like being tired vs. being relaxed, for another example see [26]). In therapeutic settings for example, a *gain* in relaxation after an intervention would ideally be accompanied by a *decrease* in sleepiness during the day to allow patients to feel both calm, but still present, attentive, and observant (corresponding to *Enjoyment* from Table 1). Thus, it is vital to be able to understand how these two states combine, interact, and affect e.g., performance, well-being, and other outcomes in different settings. Especially since overarching states (such as relaxation and sleepiness) may very well be context specific as it was already demonstrated with other constructs [26, 27]. To explore this, the present study analyses a set of 11 independent studies that all used the RSQ in various settings. Meta-analytic methods will then be able to reveal overarching trends for the relationship between sleepiness and relaxation over all studies.

**Pre-post-data.** The RSQ not only allows an efficient measurement of relaxation and sleepiness, but due to its briefness and its state-conception of the constructs, it gives the opportunity

for measuring the changes in the states and their interaction over time, e.g., over the course of an experiment. Measuring states before and after an intervention is useful and insightful for several purposes: Measuring a state before a task or intervention allows researchers not only to get a baseline measurement of the participants mental state, but also to predict outcomes based on these measurements. Accordingly, post-measurements allow conclusions to be drawn e.g., about the effect or effectiveness of an intervention. Looking closer at the dynamics and interactions of mental states before and after interventions may give insights into specific changes of feelings or perceptions of individuals due to a task or intervention. This provides researchers with the opportunity to closer examine the contextual dynamics e.g., of the relationship between sleepiness and relaxation overall.

It could already be shown that even traits (which per definition are supposed to be stable) may be sensitive to change [e.g., 28, 29]. The closer examination of states (which per definition are sensitive to change) and their dynamics and relationships may thus be insightful and enlightening. Therefore, here we also ask, how the intriguing negative relationship between sleepiness and relaxation behaves *before* and *after* interventions (that elicit a change in the subjective relaxed state) for each of the subscales of relaxation.

## Methods

### Relaxation State Questionnaire (RSQ)

The Relaxation State Questionnaire [RSQ; 23] is a 10-item questionnaire designed to assess subjective relaxation and its short term changes. Each item is written as a statement to which participants are asked to rate their agreement to on a 5-point-likert-scale (1 = do not agree at all; 5 = totally agree). The items can be divided into 4 scales: The Cardiovascular Scale, the Muscle Scale, the General Relaxation Scale, and the Sleepiness Scale. As the first three scales measure parts of relaxation, they are highly correlated ($r$ = .41, .61, and .80, respectively). Interestingly, two independent factorial analyses of the scales showed that they are also negatively correlated with the Sleepiness Scale. It was proposed that the Sleepiness Scale could therefore pose as a manipulation check e.g., an indicator for answering tendencies from participants. However, this also grants the opportunity to investigate the negative relationship between different aspects of relaxation and sleepiness and the alteration of that correlation after interventions. The RSQ has been factorially validated, has a high face validity and good reliability ($\alpha$ = 0.86), item parameters, and construct validity. Because of its efficiency it can easily be included in different studies and settings.

### Set of studies

A total of 11 experiments were included in the meta-analyses. Data collection took place over a period of three years (starting in 2021, ending in April 2023). All participants gave written informed consent before the studies and no minors were included in any of the experiments. The studies conformed to the ethical guidelines of the German Psychological Association (DGPs) and were approved by the ethics committee of Bielefeld University.

Experiments showed a great variation of settings, design, and intervention (see Table 2 for details on each experiment). Since the RSQ was mostly used in pre-post-designs, a total of 6 (3 x 2) different analyses were performed: For each of the three relaxation scales of the RSQ, two different analyses were computed. One for the *pre*-measurements of the experiments and one with the *post*-measurements. For the pre-measurements the whole sample was used for each study. For the post-measurements, the samples were divided into different groups (see Table 2), if the participants received different types of interventions. Therefore, the meta-

**Table 2. Overview of studies used for the meta-analyses.**

|  | Sample (x Sessions) | Design | Setting | Description |
|---|---|---|---|---|
| Study 1 | 6 x 10 | within | lab | 1-hour eye tracking experiment with alertness cues |
| Study 2 | 99 | survey | online | Survey after watching 45min of online Video-streaming (only in post-meta-analyses) |
| Study 3 | 95 | mixed with 3 groups | lab | Laboratory experiment with breathing exercises followed by the TMT |
| Group A | 30 | Control condition (1 min waiting) | | |
| Group B | 39 | Short relaxation condition (5 min breathing exercise) | | |
| Group C | 26 | Longer relaxation condition (11 min breathing exercise) | | |
| Study 4 | 6 x 5 | within | lab | Eye tracking experiment with Flanker Task |
| Study 5 | 61 | within | online | 30 min PMR intervention and relaxation questionnaires |
| Study 6 | 144 | within | online | 10 min PMR intervention |
| Study 7 | 109 | mixed with 2 groups | online | Stressful learning vs. relaxation experiment with relaxation and tension questionnaires |
| Group A | 57 | Relaxation condition (15 min PMR audio instruction) | | |
| Group B | 52 | Stressful condition (Vocabulary learning experiment under time pressure) | | |
| Study 8 | 5 x 10 | within | lab | Muscle tensing and relaxing exercises and TVA trials |
| Study 9 | 7 x 5 | within | lab | PMR and antisaccade task |
| t1 | 7 x 5 | First Post-Measurement after 160 trials of an antisaccade task with eye tracking | | |
| t2 | 7 x 5 | Second Post-Measurement after 15 min of PMR audio instruction | | |
| Study 10 | 49 | mixed with 3 groups | online | Online experiment with 3 group and pre-post relaxation measurements |
| Group A | 14 | Relaxing condition (15min PMR audio instruction) | | |
| Group B | 13 | Neutral condition (picture description task) | | |
| Group C | 22 | Stressful condition (Color Word Stroop under time pressure and impossible number of trials) | | |
| Study 11 | 148 | mixed with 3 groups | | PMR intervention and Mackworth-clock-task with different levels |
| Post | 147 | First measurement after PMR intervention, before Mackworth-clock-task | | |
| Group A | 48 | Mackworth-clock-task with 2 positions skipped | | |
| Group B | 59 | Mackworth-clock-task with 1 position skipped | | |
| Group C | 39 | Control (no Mackworth-clock-task) | | |

Sample denotes *N* or *n* in the subgroup, if applicable with number of sessions for each person. Within indicates a within-subjects design with repeated measures. Setting is categorized into laboratory based (lab) settings and online settings. TMT = Trail Making Test [30]. PMR = Progressive Muscle Relaxation [25]. TVA = Theory of visual attention [31]. t1 and t2 denote the different measurement time points.

analyses for the post-measurements contained 20 data subsets whereas the analyses for the pre-measurements only contained 10.

## Data analysis

The meta-analyses were performed with the *metafor* package [32]. As outcome measures, Fishers z-transformed correlation coefficient was used, because the set of studies contained studies with a rather small sample size [33, but see also 34]. Each correlation was against the Sleepiness Scale of the RSQ. Then, random effects models were fitted with maximum likelihood estimation [32] and forest plots were computed. Also, to visually control for biases in our data, funnel plots were plotted [35].

## Results

### Correlations pre-intervention

**Muscle scale.**   The pre-intervention data for the Muscle Scale showed a medium amount of heterogeneity ($I^2$ = 58.79%). The estimated correlation is β = -.26, $p$ < .001, 95% CI [-.38, -.14]. Fig 1 shows the forest plot with the corresponding correlation from each study.

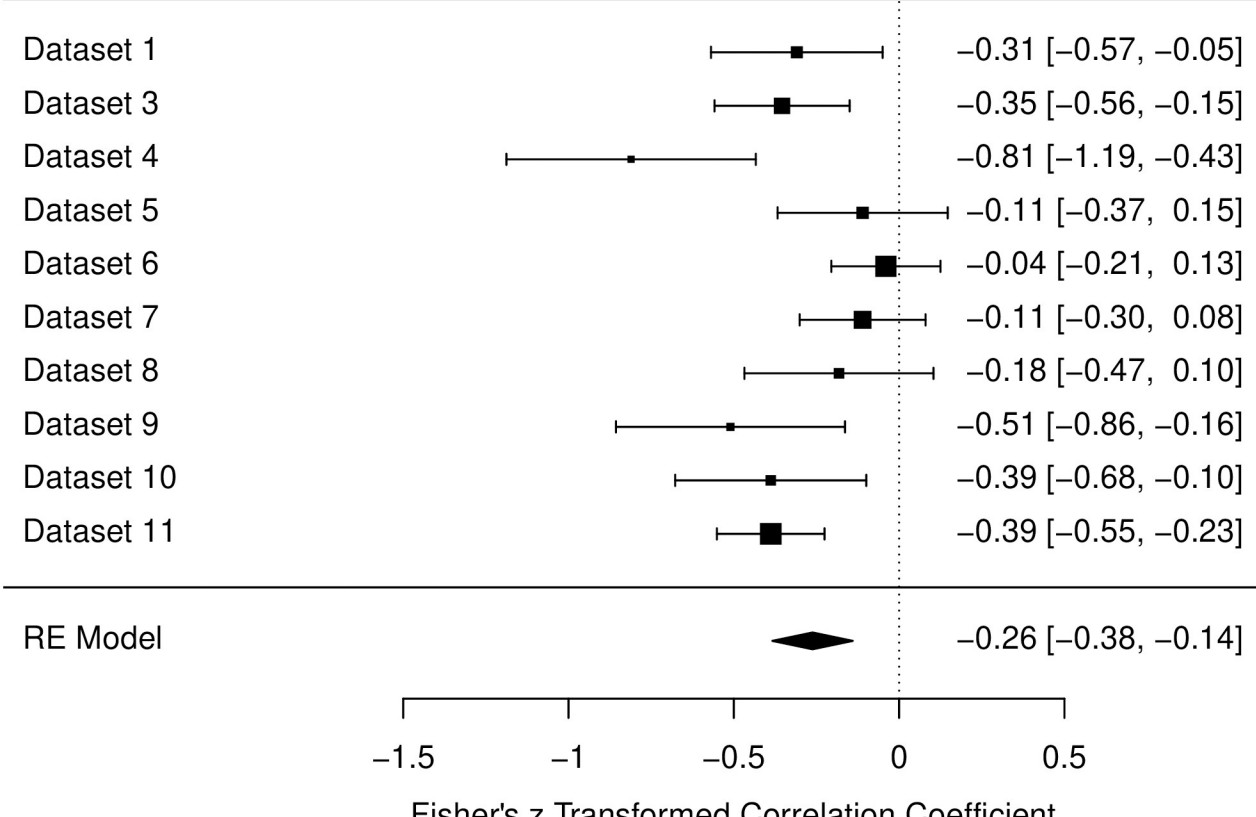

**Fig 1. Forest plot for the data of the muscle scale in the pre-intervention Measurements.** Fisher's z transformed correlation with 95-% Confidence intervals are reported for each study. Bigger black squares indicate a higher *N*. Designations of datasets are corresponding to Table 2.

**General relaxation scale.** For the General Relaxation Scale there was also a medium amount of heterogeneity in data ($I^2$ = 46.30%). The estimated correlation was similar to the one found for the Muscle Scale: β = -.27, $p$ < .001, 95% CI [-.38, -.17] (see also Fig 2).

**Cardiovascular scale.** Heterogeneity for the Cardiovascular Scale was not as high as for the other scales ($I^2$ = 35.85%). Also, the estimated correlation with sleepiness was lower for this scale (see also Fig 3), β = -.16, $p$ < .001, 95% CI [-.25, -.07].

## Correlations post-intervention

**Muscle scale.** For the post-intervention data, the heterogeneity of the Muscle Scale was $I^2$ = 16.18%. Correlations for all subsets and studies can be found in Fig 4. The estimated correlation was β = -.14, $p$ < .001, 95% CI [-.21, -.07].

**General relaxation scale.** Heterogeneity of the General Relaxation Scale data after the intervention was high, $I^2$ = 60.03%. However, the random-effects model did not reach significance (β = -.10, $p$ = .11, 95% CI [-.21, .01]) (see Fig 5).

**Cardiovascular scale.** For the post-intervention data of the Cardiovascular Scale, there was little heterogeneity, $I^2$ = 10.66%. Also, the estimated correlation did not reach significance: β = .01, $p$ = .712, 95% CI [-.07, .06]. Correlations for all subsets can be found in Fig 6.

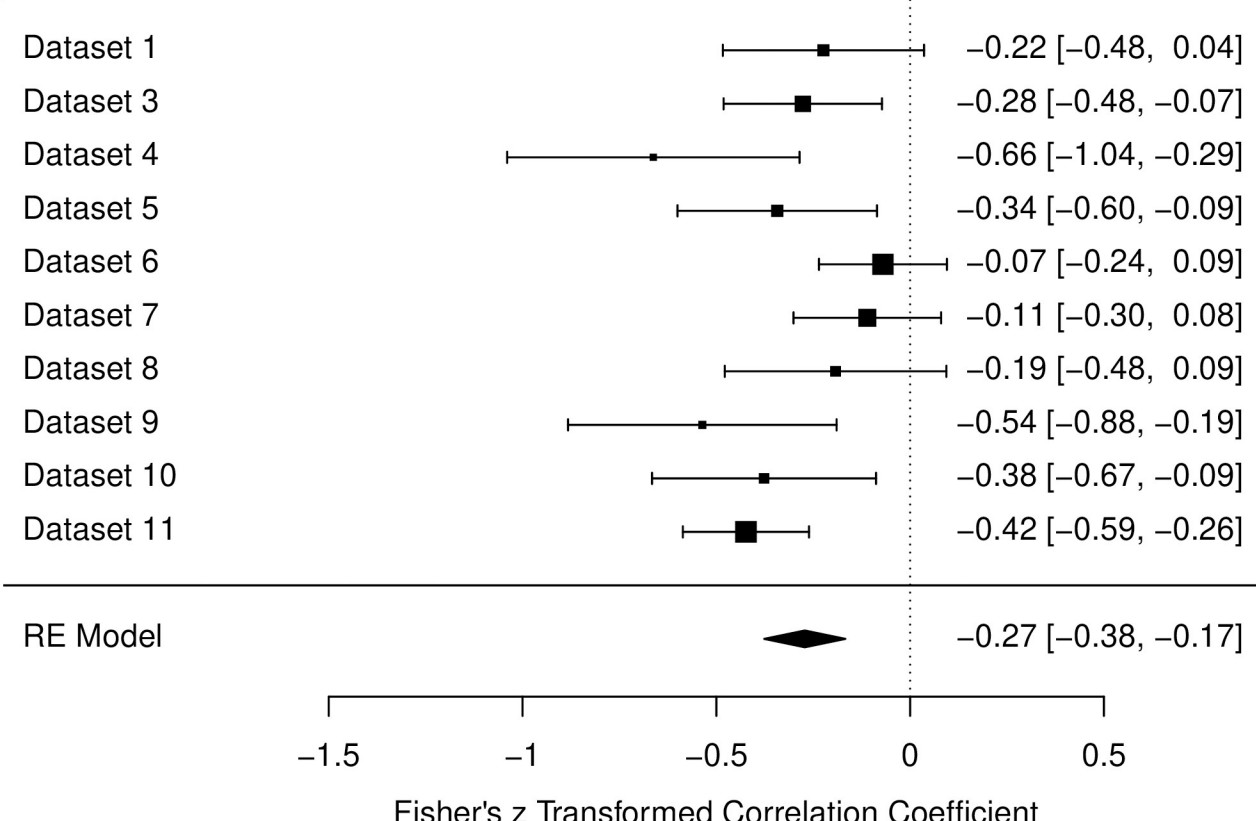

**Fig 2. Forest plot for the data of the general relaxation scale in the pre-intervention measurements.** Fisher's z transformed correlation with 95-% Confidence intervals are reported for each study. Bigger black squares indicate a higher *N*. Designations of datasets are corresponding to Table 2.

### Further analyses

To control for biases with the experiments (e.g., hidden selection effects), funnel plots were computed for each random-effects model. All plots can be found in the supporting information. Also, given the broad variation of studies, some aspects of the design and setting were categorized and their influence on the correlation was investigated. The factors setting (online vs. lab-based), design (repeated-measures vs. not), and type of intervention (relaxing intervention vs. not) were analyzed using *t*-tests and the Bayes Factor [36]. The Bayes Factor gives information about how strongly the two hypotheses (alternative vs. null hypothesis) support the existing data and can be seen as an alternative to the *p* value [37, 38]. Following guidelines for categorizing Bayes Factors [BF10; 39], the resulting outcomes can be quantified as follows: $BF_{10}$ of 1 = equal support for the null and the alternative hypothesis, 1 to 3 = weak evidence in favor of the alternative hypothesis, and 3 to 10 = moderate evidence in favor of the alternative hypothesis. However, except for one comparison (see Table 3) no significance difference was found for either setting, design, or type of intervention. The Bayes Factor mostly supports these findings, indicating moderate evidence only for the significant comparison. Furthermore, two more BF were just above 1, indicating weak evidence for the alternative hypothesis. Table 3 shows exemplary the data for the setting-comparisons. The other comparisons can be found in S1 and S2 Tables.

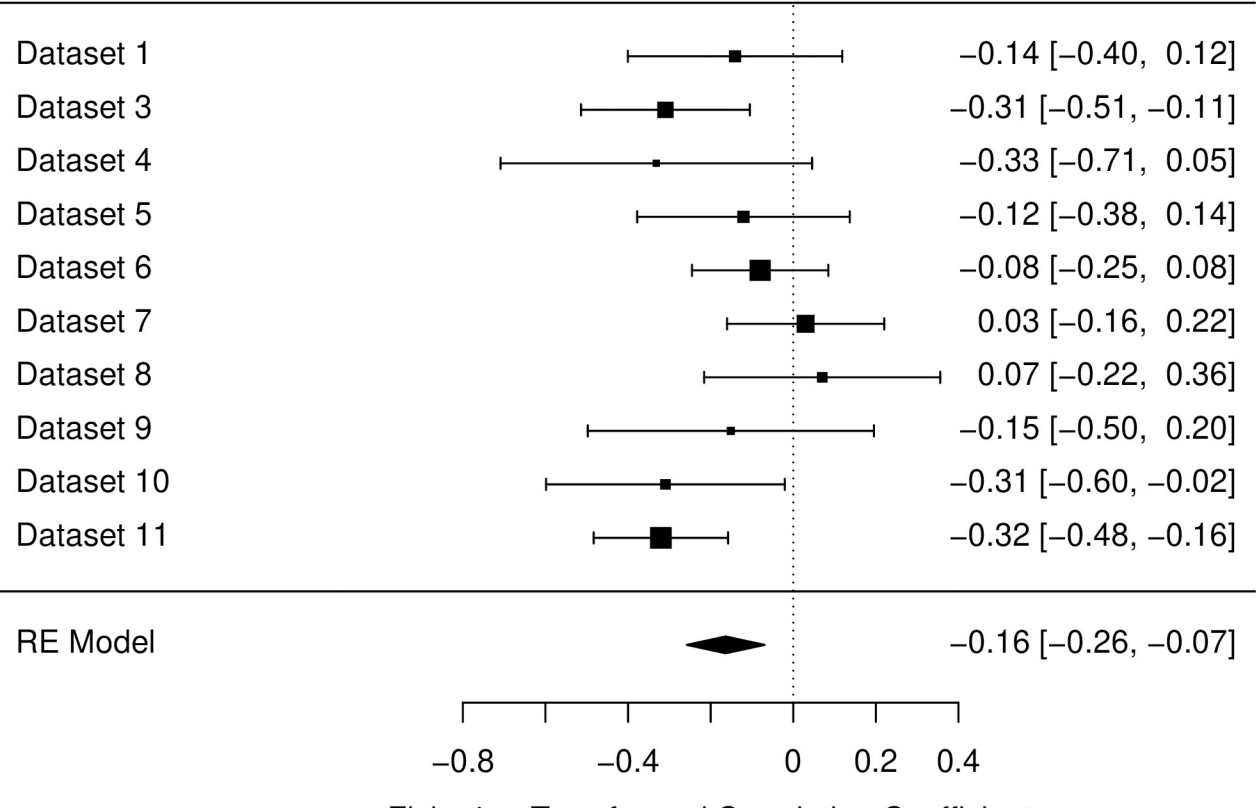

**Fig 3. Forest plot for the data of the cardiovascular scale in the pre-intervention measurements.** Fisher's z transformed correlation with 95-% Confidence intervals are reported for each study. Bigger black squares indicate a higher *N*. Designations of datasets are corresponding to Table 2.

## Discussion

The relationship between relaxation and sleepiness has been in the focus of research for many decades [19]. While the idea of a general dimension of arousal or activation can be useful [40], different research suggests that there are indeed two distinct dimensions of arousal, that seem to be negatively correlated [22, 41]. To extend the existing evidence and applications, [e.g., 42–44], this paper makes use of a broad variety of datasets to further investigate the relationship between relaxation and sleepiness and their dynamics before and after interventions. Using the Relaxation State Questionnaire [RSQ; 23], an economic and reliable tool that assess both short-term subjective relaxation and sleepiness, and their relationship.

The data collected for measurements before any form of intervention, included 10 different studies including both laboratory based and online studies. For all three scales of relaxation (muscle related aspects, general relaxation, and cardiovascular related aspects) a significant negative correlation with sleepiness could be found over all data sets (*r* = -.26, -.27, and -.16, respectively). Notably, almost every study produced a descriptive negative correlation, besides Studies 7 and 8 for the Cardiovascular Scale. Therefore, in over 93% of instances, the negative correlation was found. It is also worth mentioning that in the process of constructing the RSQ, the Cardiovascular scale displayed less strong item parameters and factor loadings [23]. Hence, the slightly lower correlation of *r* = -.16 and the non-negative correlations for Studies 7 and 8 in the Cardiovascular Scale, may stem from limitations regarding the measurement, not

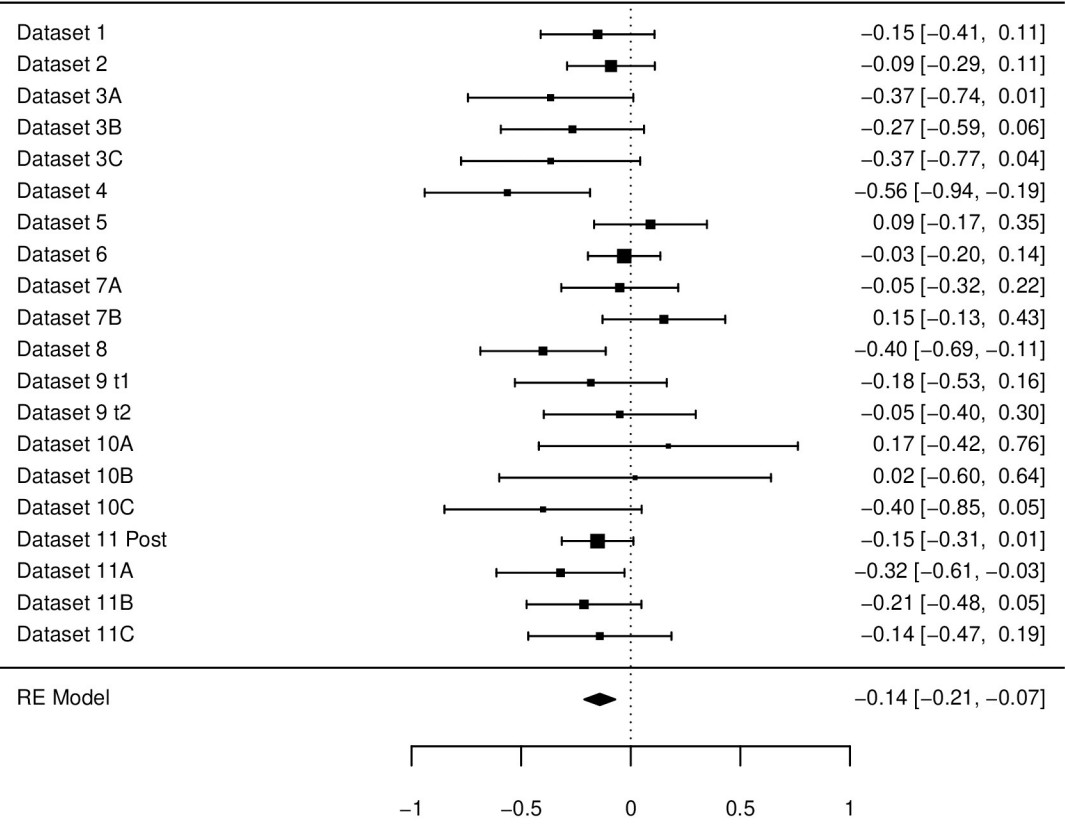

**Fig 4. Forest plot for the data of the muscle scale in the post-intervention measurements.** Fisher's z transformed correlation with 95-% Confidence intervals are reported for each study. Bigger black squares indicate a higher *N*. Designations of datasets are corresponding to Table 2.

from an actual absence of a (high) negative correlation. Overall the three analyses from the pre-intervention datasets therefore confirm findings indicating that sleepiness and relaxation are two distinct dimensions and are negatively correlated [41, 45]. Our findings suggest that this relationship is consistent over different settings and environments.

The data collected for the measurements *after* an intervention, included 20 data subsets with a broad variety of interventions and settings. The descriptive correlations for the three relaxation scales were *r* = -.14, -.10, and -.01. However, only the first correlation reached significance. All correlations are also less negative than in the pre-measurement analyses. Also, in the forest plots one can find several data sets that actually produced (significant) *positive* correlations (e.g., Fig 6, Dataset 10A). In total 21 datasets were descriptively positive, meaning that only 65% percent of the datasets showed a negative correlation. Again, the Cardiovascular Scale had the least negative correlation and with 11 positive and 9 negative datasets, the most mixed results. There are several possible explanations for these findings. First, it is possible that because the data, the conditions, and the subgroups used in the analyses were very diverse, too many variables and therefore too much noise was introduced into the datasets (see also discussion of the further analyses below). Secondly, on a more general note, one could also raise the question on how participants are able to obtain meta-cognitive knowledge about their own subjective relaxation and their sleepiness. It is widely known that subjective and objective

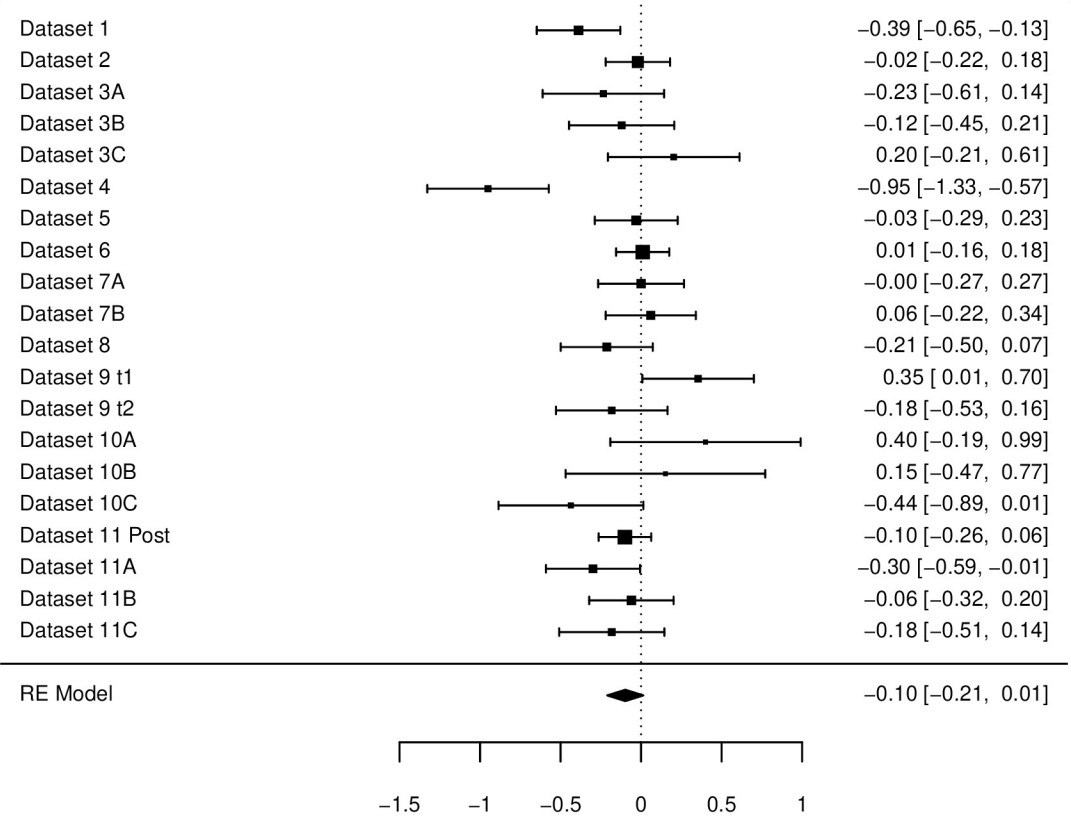

**Fig 5. Forest plot for the data of the general relaxation scale in the post-intervention measurements.** Fisher's z transformed correlation with 95-% Confidence intervals are reported for each study. Bigger black squares indicate a higher *N*. Designations of datasets are corresponding to Table 2.

measures may differ in their outcomes and are therefore often both assessed to capture the whole picture of a construct [e.g., 3, 9, 46]. While subjective measures have the advantage of giving insight in the participants personal assessment and feelings, they are also prone to biases such as demanding characteristics or tendencies of certain self-presentations. In the present studies, it is plausible that participants answered the RSQ before an intervention rather honestly, however *after* an intervention participants answered in the direction, they thought the experiment was headed (e.g., claiming to be more relaxed after a relaxation exercise). For the sleepiness, however, it may be a lot less clear, what the 'right' answer after an intervention could be. Since both sleepiness and relaxation were once thought to be one dimension and are both associated with low arousal [19], participants might have rated both categories similar. For example, after completing a relaxation exercise, they might have thought that they were supposed to be more relaxed *and* sleepier afterwards. This would also be in line with the original recommendation of the RSQ, to use the sleepiness scale as a manipulation check for answering tendencies, since participants may not be able to distinguish between the aspects of relaxation and sleepiness [23]. To control for effects like these, objective measurements (e.g., heart rate, blood pressure) may need to be used in future studies to gain further insight in the objective aspects of relaxation and sleepiness accompanying the subjective ratings. Thirdly, it seems plausible that the found correlation is indeed accurate and not distorted by the data or

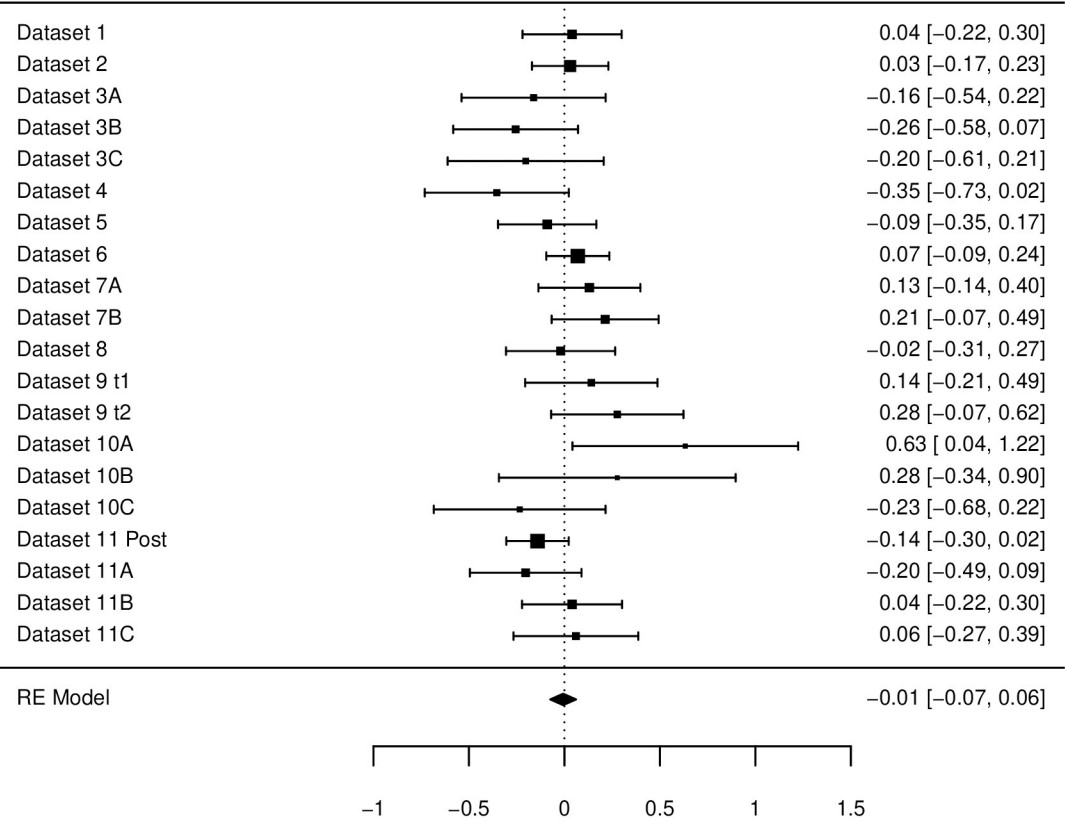

**Fig 6. Forest plot for the data of the cardiovascular scale in the post-intervention measurements.** Fisher's z transformed correlation with 95-% Confidence intervals are reported for each study. Bigger black squares indicate a higher *N*. Designations of datasets are corresponding to Table 2.

the participants answering tendencies. The difference between the pre-measurements and the post-measurements is the intervention taking place in between. Depending on the type of intervention, this may affect relaxation and sleepiness quite differently. There are some studies that indicate that certain interventions can lead to different outcomes on the two dimensions [e.g., 47]. Not only are both energetic and tense arousal prone to change differently, also their dynamic may change depending on the situation or intervention [48, 49]. One idea by Thayer is that with moderate activation conditions both tense and energetic arousal behave similarly,

**Table 3. Mean correlations for each scale depending on setting.**

|  | Pre-intervention | | | | Post-intervention | | | |
|---|---|---|---|---|---|---|---|---|
|  | Online (*n* = 5) | Lab (*n* = 5) | *p* | $BF_{10}$ | Online (*n* = 12) | Lab (*n* = 8) | *p* | $BF_{10}$ |
|  | *M* (*SD*) | *M* (*SD*) | | | *M* (*SD*) | *M* (*SD*) | | |
| Muscle Scale | -.2 (.16) | -.39 (.19) | .118 | 1.14 | -.08 (.17) | -.28 (.15) | **.013** | 3.95 |
| General Relaxation Scale | -.25 (.15) | -.35 (.17) | .381 | 0.64 | -.04 (.2) | -.16 (.33) | .369 | 0.58 |
| Cardio-vascular Scale | -.16 (.15) | -.17 (.16) | .904 | 0.49 | .06 (.22) | -.06 (.21) | .221 | 0.69 |

*n* = number of studies in this category; *p* values in bold indicate significance

however when one dimension is highly activated (e.g., extreme stress or pain for tense arousal and exercise or drugs for energetic arousal), this dimension "takes over" and the other dimension does not rise [49]. It could be very interesting for future experiments to manipulate the level of tense and/or energetic arousal in different stages (low–medium–high) and measure with the RSQ both relaxation and sleepiness of the participants. Thus, a mapping of the two states and their relationship depending on the different arousal levels may be possible. Lastly, the further analyses of classification of the setting, the type of intervention, and the design of the studies did not reveal significant impacts on the found correlations. Only the difference in laboratory-based vs. online studies in the post-datasets for the Muscle Scale was significant, with laboratory-based studies yielding more negative correlations ($r$ = -.28) that online-based studies ($r$ = -.08). However, due to the large amount of test, one significant finding seems not surprising and should be interpreted cautiously. Nonetheless, this finding calls for further substantiation by follow-up studies, because it may suggest that the relationship between relaxation and sleepiness depends on the context, and that laboratory and online assessments might influence the relationship (e.g. by modulating how the variance of test scores is dominated by experimental and individual influences, [cf. 50, 51]).

Another possible explanation for these findings regards the broad variety of studies, conditions, and datasets. While it was the goal to obtain different datasets similar to 'traditional' meta-analyses, this also comes with the cost that the number of completely different settings and designs generates a lot of noise that may pollute the data. Classifications made for testing the different settings and types of interventions for example were hence very rough. Relaxation exercises e.g. contained both breathing exercises of different length and progressive muscle relaxation exercises [24, 25] of different length and quality. It has been shown in several studies that different relaxation exercises produce different effects and outcomes as they have distinct principles of operation [e.g., 52, 53]. Datasets in the non-relaxing condition were even more divers and ranged from waiting control conditions, to stressful exercises, to neutral tasks. The classification of *relaxing vs. not* could hence not capture all aspects within the data. Therefore, a number of more similar studies using the RSQ could shed more light on possible effects of a certain relaxation exercise or setting on the correlation between relaxation and sleepiness.

## Conclusion

While relaxation and sleepiness are both associated with low arousal levels, the two constructs are two distinct dimensions. Using a broad variety of datasets, meta-analytical methods, and the Relaxation State Questionnaire [23], we could share some light on the relationship between the constructs that are based on two distinct arousal dimensions. While the data before a task or intervention suggests a consistency of the negative correlation between sleepiness and relaxation, the post-data showed a very mixed and inconsistent relationship between the two constructs. Thus, the intricated interplay between the two dimensions may be dependent on a variety of factors and may therefore change over short periods of time, even over the intercourse of an experiment. Even though more research is necessary to unravel the details of these changes, the present datasets and analyses may serve as a solid foundation for future explorations.

## Supporting information

**S1 Fig. Funnel plot muscle scale pre-intervention.** Funnel Plot of the used datasets. Each point plotted represents the weighted z-transformed correlation of one dataset described in Table 2. The vertical dotted line represents the estimated correlation, as reported in Fig 1. The white triangle represents the region in which 95% of the data points should lie in absence of a

selection bias.
(PDF)

**S2 Fig. Funnel plot general relaxation scale pre-intervention.** Funnel Plot of the used data-sets. Each point plotted represents the weighted z-transformed correlation of one dataset described in Table 2. The vertical dotted line represents the estimated correlation, as reported in Fig 2. The white triangle represents the region in which 95% of the data points should lie in absence of a selection bias.
(PDF)

**S3 Fig. Funnel plot cardiovascular scale pre-intervention.** Funnel Plot of the used datasets. Each point plotted represents the weighted z-transformed correlation of one dataset described in Table 2. The vertical dotted line represents the estimated correlation, as reported in Fig 3. The white triangle represents the region in which 95% of the data points should lie in absence of a selection bias.
(PDF)

**S4 Fig. Funnel plot muscle scale post-intervention.** Funnel Plot of the used datasets. Each point plotted represents the weighted z-transformed correlation of one dataset described in Table 2. The vertical dotted line represents the estimated correlation, as reported in Fig 4. The white triangle represents the region in which 95% of the data points should lie in absence of a selection bias.
(PDF)

**S5 Fig. Funnel plot general relaxation scale post-intervention.** Funnel Plot of the used data-sets. Each point plotted represents the weighted z-transformed correlation of one dataset described in Table 2. The vertical dotted line represents the estimated correlation, as reported in Fig 5. The white triangle represents the region in which 95% of the data points should lie in absence of a selection bias.
(PDF)

**S6 Fig. Funnel plot cardiovascular scale post-intervention.** Funnel Plot of the used datasets. Each point plotted represents the weighted z-transformed correlation of one dataset described in Table 2. The vertical dotted line represents the estimated correlation, as reported in Fig 6. The white triangle represents the region in which 95% of the data points should lie in absence of a selection bias.
(PDF)

**S1 Table. Mean correlations for each scale depending on intervention type.** $n$ = number of studies in this category; $p$ values in bold indicate significance.
(PDF)

**S2 Table. Mean correlations for each scale depending on the design.** $n$ = number of studies in this category; $p$ values in bold indicate significance.
(PDF)

## Author Contributions

**Conceptualization:** Sarah Steghaus, Christian H. Poth.

**Data curation:** Sarah Steghaus.

**Formal analysis:** Sarah Steghaus.

**Investigation:** Sarah Steghaus.

**Methodology:** Sarah Steghaus.

**Project administration:** Christian H. Poth.

**Resources:** Sarah Steghaus.

**Supervision:** Christian H. Poth.

**Visualization:** Sarah Steghaus.

**Writing – original draft:** Sarah Steghaus.

**Writing – review & editing:** Sarah Steghaus, Christian H. Poth.

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
