## [Decision Letter · Decision Letter 0]

15 Jul 2024

PONE-D-24-21576Feeling tired vs. feeling relaxed: Two faces of low physical arousalPLOS ONE

Dear Dr. Steghaus,

Thank you for submitting your manuscript to PLOS ONE. After careful consideration, we feel that it has merit but does not fully meet PLOS ONE’s publication criteria as it currently stands. Therefore, we invite you to submit a revised version of the manuscript that addresses the points raised during the review process. **Editor comments. **I was able to appoint only one reviewer to comment on your manuscript. I asked a large number of potential referees, but so far without success. It is currently very difficult to find reviewers, and this seems to be a widespread issue. Therefore, I decided to step in this time and act as the second reviewer. We need a second referee for the final phase of the manuscript, but for now, my input should suffice to maintain the workflow and to give you some feedback for your work. Both R1 and my own reading suggest that this manuscript has the potential for significant impact. The manuscript has many strengths, particularly in the theoretical domain. It addresses a very general and relevant problem for nearly every field, differentiating theoretically between similar-sounding concepts. R1 has some issues that should be addressed in a proper revision. I suggest providing a response letter in the revision that addresses all issues in a point-by-point reply.

We look forward to receiving your revised manuscript.

Kind regards,

Michael B. Steinborn, PhD

Section Editor

PLOS ONE

Journal Requirements:

**Additional Editor Comments:**

R1 raises valid and constructive points aimed at enhancing clarity, coherence, and significance of your manuscript. R1 raises one point that concerns the motivation of the study and how it is placed within existing knowledge. R1 argues that the study confirms known relationships but the increment offered by the study should be justified more. Additionally, R1 suggests integrating the research question more thoroughly throughout the manuscript and discussing the implications of the findings more theoretically, specifically regarding the concepts of feeling tired versus feeling relaxed. This means that revising the manuscript should involve presenting a stronger theoretical foundation in the introduction that naturally converges to the research question, and in the discussion, elaborating on how the present findings of multiple studies change the theoretical landscape. I agree with this point, as it would indeed enhance the impact of the present study. In the following, I provide comments regarding various issues I find relevant for the revision. Additionally, I will address and discuss comments from R1 to clarify their meaning and how to handle them. I want to emphasise that my comments are intended to improve the manuscript, not to criticise your work. I do not expect you to agree with all of my points; in other words, I do not claim that my opinion is always correct.

(-1-) conformation versus novelty

R1 raises the issue of whether the results confirm previous knowledge or provide novel findings capable of reshaping the theoretical landscape in the field. They argue that the present meta-analysis confirms a negative correlation between relaxation and sleepiness, but it is unclear what new insights this study adds to existing research. R1 suggests the manuscript should explain the importance of understanding the relationship between relaxation and sleepiness and why a meta-analysis is indicated. While I completely agree with these points, I want to note that systematically verifying findings for consistency is per se important and valuable in empirical research fields. This means that "novelty" should not be misconstrued as "discovery", especially in a field of behavioural research that is prone to false-positive results. Therefore, in my view, a lack of novelty is not an issue simply because a correlation has been reported previously. It is crucial to recognise that many findings in the correlational domain of states are often inconsistent. Demonstrating consistency, particularly in relation to specific variables or experimental contexts, is in my view, both significant and valuable.

(-2-) introduction

R1 finds that the research question about the relationship between sleepiness and relaxation is not well-integrated into the manuscript at present, and I completely agree with this point. I have some comments on how to handle this issue in a revision.

(--) Energetic Arousal (EA) vs. Tense Arousal (TA)

Thayer (1990) conceptualised energetic arousal (EA) and tense arousal (TA) as independent yet interactive components of reportable mental states in everyday situations. EA relates to feelings of vigour and vitality, while TA pertains to feelings of tension and nervousness, and moreover, these dimensions can combine in various ways to yield different mixtures of mental states. For example, high Energetic Arousal (EA+) and High Tense Arousal (TA+) is a state that corresponds to something we would call "challenge", this combination is often experienced in situations requiring high effort and involvement, such as competitive sports or high-stakes tasks, with the individual feeling energetic and engaged but also experiences a degree of stress or pressure. On the other hand, EA+ and TA- refers to a state that corresponds to excitement or pleasure, where individual feel lively and positive but in a relaxed way without the accompanying stress or extreme urgency to engage in an activity. This is typically observed in enjoyable activities where one is fully engaged but relaxed, such as playing a favourite sport for fun or engaging in a stimulating hobby. Further, EA- and TA+ is a state that corresponds to anxious agitation, this combination is marked by feelings of tension and worry without the counterbalance of energy. It is commonly experienced in situations perceived as threatening or overwhelming. Finally, EA- and TA- is a state that corresonds to apathy, which is a a state of low energy and low tension at the same time, often characterised by a lack of motivation and interest.

(--) Motivating the present study

While state questionnaires are often used in experimental settings, the concepts underlying these states are often not explicitly distinguished, as they are merely applied (i.e., this is partly because most researchers do not extensively investigate fields they consider peripheral to their own). However, this leads to the common issue of conflating similar-sounding constructs that have entirely different meanings (e.g., being tired vs. being relaxed), while treating seemingly dissimilar constructs (e.g., satiation vs. ego-depletion) as distinct (see Schumann et al., 2022, doi:10.3389/fpsyg.2022.867978, see Table 2). Researchers are indeed capable of distinguishing these if they engage with the specialised literature, but they often do not, as we have cognitive biases and limited resources. This exactly is the reason why specific knowledge from expert domains does typically not adequately permeate into other areas. Therefore, what is well-known and unremarkable to specialists can be a surprising and valuable insight to other researchers. It is therefore crucial to differentiate between what is "already known" by highly-specialised experts and what is common knowledge or "common sense" (what everyone knows) by everyone in a community of researchers covering various fields.

(--) precise research question

The pertinent question for virtually all fields of experimental psychological research (be it social, cognitive, or clinical) is to know how these state combinations affect performance in concrete settings, beyond everyday situations. To know means to understand systematically and coherently, not merely to be aware of one or some previous studies that have shown something (e.g., a correlational relationship) in a specific context using a specific sample of individuals. In experimental psychology, we do not speak of knowledge when referring to a single case but only when understanding a general principle. Therefore, it is crucial to examine how pre-test mental states relate to performance levels, how these states change over the course of a performance situation, and how they correlate with each other (e.g., before and after testing). I concur with the authors that it is vital to provide consistency and context variations across a variety of experiments. It is fundamentally different to play Tetris, to perform a simple-RT task, or engage in a low-event-rate vigilance task where nothing happens for extended periods. Moreover, it matters whether the performance situation is structured or motivated by goal settings, e.g., if mini-breaks are given or if mind-wandering, which equates to unregistered breaks, is possible (see: doi:10.3389/fpsyg.2022.867978; doi:10.3758/s13414-023-02803-4). Therefore, understanding these relationships is essential for elucidating the impact of mental states on performance outcomes. Conducting a systematic meta-analysis is an ideal approach to investigate this systematically.

(-3-) Pre- and Post-Intervention Data:

R1 notes that the value of using both pre- and post-intervention data is not well-articulated, which I agree and thereby would like to comment further on this aspect:

Considering pre-test to post-test intervals when conducting experiments is crucial for several reasons. Firstly, measuring states explicitly before and after the demand allows researchers to determine how mental states predict upcoming performance requirements. Secondly, it is essential to understand whether and how self-reported states change during performance demands, as this provides an indication of the costs or benefits these demands may generate on feelings. Thirdly, it provides insights into the contextual dynamics of the relationships between state dimensions, specifically how pre-test relationships might change correlatively over the course of an experiment. This is important because researchers often assume that traits are static and unchangeable, and not seldom, they implicitly extend this assumption to states, despite their inherent variability. To name one example of why this is absolutely crucial, Kärtner et al. (2021, doi:10.1038/s41598-021-81446-7) have shown that even stable traits are sensitive to change in some contexts. This suggests that not only states but even traits assumed to be stable by definition should be measured multiple times, or at least should be checked for variability in specific contexts. If even traits are somewhat variable and not static under some circumstances, this would clearly indicate that multiple measurements are, if not necessary, worth considering. This demonstrates that even firmly held beliefs can manifest empirically in completely different ways.

Reviewers' comments:

Reviewer's Responses to Questions

**Comments to the Author**

1. Is the manuscript technically sound, and do the data support the conclusions?

Reviewer #1: Yes

2. Has the statistical analysis been performed appropriately and rigorously? 

Reviewer #1: Yes

3. Have the authors made all data underlying the findings in their manuscript fully available?

Reviewer #1: Yes

4. Is the manuscript presented in an intelligible fashion and written in standard English?

Reviewer #1: Yes

5. Review Comments to the Author

Reviewer #1: Using data from 11 experiments that used the Relaxation State Questionnaire (RSQ), the authors conducted a meta-analysis to investigate the relationship between relaxation and sleepiness. The authors found that the three aspects of relaxation as measured by the RSQ (cardiovascular, muscle, or general relaxation) were negatively correlated with the sleepiness factor of the RSQ when measured before any relaxation interventions. After relaxation interventions, however, the negative correlations between the three relaxation scales and the sleepiness scale were not only weaker, but also only the correlation with the muscle scale was significant. Further analyses which explored possible moderators did not report any significant effects.

This meta-analysis provides further evidence of the distinction between the low arousal states of relaxation and sleepiness, and while limited by its use of only the RSQ, uses a large, diverse dataset, and all the authors’ data and analytical scripts are openly available. However, my main point of revision is that, despite the relationship between the two states being described as “complex”, it is unclear what additional information has been learned from this meta-analysis beyond confirming the negative correlations observed in the original RSQ paper. Indeed, the way the manuscript is currently set up, it comes across that the negative relationship between relaxation and sleepiness has already been established, and why that relationship matters and why a meta-analysis is needed (for researchers, practitioners, or other audiences) is not strongly explained. Additionally, it is not clear to me what value has been added by using both the pre- and post-intervention data. The authors provide an intriguing research question at the end of the introduction (p. 10): “here we ask, how this intriguing negative relationship between sleepiness and relaxation behaves before and after interventions (that elicit a change in the subjective relaxed state) for each of the subscales of relaxation.” Yet this research question is absent from the abstract and conclusion, and when it is discussed, the authors focus on explaining the findings away rather than discussing why these changes, if real, matter and what they could tell us about the relationship between relaxation and sleepiness. In sum, I do not have strong objections to this manuscript's publication, but I believe this manuscript’s value to the academic literature would be greatly strengthened by the authors reshaping the manuscript to emphasize the importance of their findings.

Minor points of revision:

• Is Table 2 (p. 10, paragraph 2 under “Set of Studies”) the correct table? Table 2 appears to be one of the tables discussed in the “Further analyses” (which are otherwise in the Supporting Information); however, in the text Table 2 is described as information on the post-measurement samples, which seems to be a reference to Table 1.

• Although it can be figured out from the results, it would be helpful to explicitly note in either the “Set of Studies” (p. 10) or the “Data Analysis” section (p. 14) that the correlations are against the sleepiness scale of the RSQ. At the moment, it only says in the “Set of Studies” section that it will be for “each of the three relaxation scales of the RSQ.”

• The section for “Further analyses” (p. 16) discusses the lack of significant differences but does not discuss the results from the Bayes analyses, despite dedicating two sentences to explaining how to interpret Bayes Factors. A simple sentence on the weak BFs found would be sufficient to help readers who do not look at the Supporting Information.

• It would be helpful for readers to have titles for the tables and captions for the figures included in the Supporting Information. Apologies if these already exist; they aren’t present in the reviewer version.

6. PLOS authors have the option to publish the peer review history of their article (what does this mean?). If published, this will include your full peer review and any attached files.

Reviewer #1: No

---

## [Author Response · Author response to Decision Letter 0]

21 Aug 2024

Review Comments to the Author (Reviewer #1): 

Using data from 11 experiments that used the Relaxation State Questionnaire (RSQ), the authors conducted a meta-analysis to investigate the relationship between relaxation and sleepiness. The authors found that the three aspects of relaxation as measured by the RSQ (cardiovascular, muscle, or general relaxation) were negatively correlated with the sleepiness factor of the RSQ when measured before any relaxation interventions. After relaxation interventions, however, the negative correlations between the three relaxation scales and the sleepiness scale were not only weaker, but also only the correlation with the muscle scale was significant. Further analyses which explored possible moderators did not report any significant effects.

We would like to thank the reviewer for their helpful and constructive comments. Please find our responses to the comments below. We also uploaded a file with all the comments and our responses, where the original comments are repeated (in black), and our responses are written in italics and blue font.

This meta-analysis provides further evidence of the distinction between the low arousal states of relaxation and sleepiness, and while limited by its use of only the RSQ, uses a large, diverse dataset, and all the authors’ data and analytical scripts are openly available. However, my main point of revision is that, despite the relationship between the two states being described as “complex”, it is unclear what additional information has been learned from this meta-analysis beyond confirming the negative correlations observed in the original RSQ paper. Indeed, the way the manuscript is currently set up, it comes across that the negative relationship between relaxation and sleepiness has already been established, and why that relationship matters and why a meta-analysis is needed (for researchers, practitioners, or other audiences) is not strongly explained. 

Thank you for your valuable comment. To shed more light on the possible dynamics and interaction between sleepiness and relaxation, added the following information to our introduction (page 3, lines 23 ff.): 

Furthermore, the two dimensions can interact in different ways to produce various combinations of mental states [19]. EA and TA may be categorized into high (+) and low (-) states, allowing four different combinations of EA and TA. Table 1 shows a fourfold table with all possible combinations, giving everyday situations where they may occur. 

Table 1. Fourfold table With Arousal Combinations

 TA+ TA-

EA+ Challenge or high-stakes scenario

Individuals are very engaged, involved and energetic, but also under a certain degree of pressure, urgency or stress

Examples: Public speaking, important exam, competitive sports

 Enjoyment or pleasure

Individuals are energetic and positive, but relaxed and calm at the same time

Examples: Non-competitive sports, playing games, pursuing a hobby

EA- Anxiousness

Individuals are tense, nervous and/ or worrying, but at the same time without energy

Examples: Threatening or overwhelming situations Passivity

Individuals are experiencing a lack of energy and enthusiasm while also experience no tension 

Examples: Lack of motivation, apathy, indifference

TA+ = high tense arousal. TA- = low tense arousal. EA+ = high energetic arousal. EA- = low energetic arousal.

We also added a new section to the introduction (“Research Questions and Objectives”) to clarify our motivation for this study (p. 4, lines 31 ff.):

“Research Questions and Objectives

Further understanding the complex relationship between sleepiness and the facets of relaxation can impact both the theoretical understanding of the constructs and have a variety of practical implications, e.g., concerning stress management or well-being. While state questionnaires (and other measures) are frequently used in research, especially in experimental settings, the underlying concepts are often not clearly distinguished. This may lead to confusion between similar-sounding terms with different meanings (like being tired vs. being relaxed, for another example see [26]). In therapeutic settings for example, a gain in relaxation after an intervention would ideally be accompanied by a decrease in sleepiness during the day to allow patients to feel both calm, but still present, attentive, and observant (corresponding to Enjoyment from Table 1). Thus, it is vital to be able to understand how these two states combine, interact, and affect e.g., performance, well-being, and other outcomes in different settings. Especially since overarching states (such as relaxation and sleepiness) may very well be context specific as it was already demonstrated with other constructs [26,27]. To explore this, the present study analyses a set of 11 independent studies that all used the RSQ in various settings. Meta-analytic methods will then be able to reveal overarching trends for the relationship between sleepiness and relaxation over all studies.”

Additionally, it is not clear to me what value has been added by using both the pre- and post-intervention data. The authors provide an intriguing research question at the end of the introduction (p. 10): “here we ask, how this intriguing negative relationship between sleepiness and relaxation behaves before and after interventions (that elicit a change in the subjective relaxed state) for each of the subscales of relaxation.” Yet this research question is absent from the abstract and conclusion, and when it is discussed, the authors focus on explaining the findings away rather than discussing why these changes, if real, matter and what they could tell us about the relationship between relaxation and sleepiness. 

Thank you again, for your very helpful comment. We now added further information on the pre-post distinction in the introduction by adding a new section to our “Research Question and Objectives”: 

“Pre-Post-Data 

 The RSQ not only allows an efficient measurement of relaxation and sleepiness, but due to its briefness and its state-conception of the constructs, it gives the opportunity for measuring the changes in the states and their interaction over time, e.g., over the course of an experiment. Measuring states before and after an intervention is useful and insightful for several purposes: Measuring a state before a task or intervention allows researchers not only to get a baseline measurement of the participants mental state, but also to predict outcomes based on these measurements. Accordingly, post-measurements allow conclusions to be drawn e.g., about the effect or effectiveness of an intervention. Looking closer at the dynamics and interactions of mental states before and after interventions may give insights into specific changes of feelings or perceptions of individuals due to a task or intervention. This provides researchers with the opportunity to closer examine the contextual dynamics e.g., of the relationship between sleepiness and relaxation overall. 

 It could already be shown that even traits (which per definition are supposed to be stable) may be sensitive to change [e.g., 28,29]. The closer examination of states (which per definition are sensitive to change) and their dynamics and relationships may thus be insightful and enlightening. Therefore, here we also ask, how the intriguing negative relationship between sleepiness and relaxation behaves before and after interventions (that elicit a change in the subjective relaxed state) for each of the subscales of relaxation.”

(p. 5, lines 5 ff.) 

Additionally, we added some sentences for further clarification in the discussion: Firstly, in the summary at the beginning of the discussion (p. 12, line 10): 

“[…] this paper makes use of a broad variety of datasets to further investigate the relationship between relaxation and sleepiness and their dynamics before and after interventions.”

Secondly, as a short summary regarding the pre-data findings (p. 12, lines 27 ff.):

“Our findings suggest that this relationship is consistent over different settings and environments.”

And thirdly, as the reviewer suggested, we addressed the topic in our conclusion (p. 14, lines 11 ff.): 

“While the data before a task or intervention suggests a consistency of the negative correlation between sleepiness and relaxation, the post-data showed a very mixed and inconsistent relationship between the two constructs. Thus, the intricated interplay between the two dimensions may be dependent on a variety of factors […]”

We also re-wrote parts of our abstract (p. 2, lines 5 ff. and 15 ff.):

“Relaxation and sleepiness are both assumed to be states of low physiological arousal and negatively correlated. However, it is still unclear how consistent this negative relationship is across different settings and whether it changes before and after an intervention.”

[…]

“For the post-intervention relationship, this negative correlation could not be consistently found. This indicates that there are aspects of the experimental setting or intervention that introduce changes in the dynamics of the relationship of the two constructs.”

In sum, I do not have strong objections to this manuscript's publication, but I believe this manuscript’s value to the academic literature would be greatly strengthened by the authors reshaping the manuscript to emphasize the importance of their findings.

Thank you again for your helpful and constructive feedback! 

Minor points of revision:

• Is Table 2 (p. 10, paragraph 2 under “Set of Studies”) the correct table? Table 2 appears to be one of the tables discussed in the “Further analyses” (which are otherwise in the Supporting Information); however, in the text Table 2 is described as information on the post-measurement samples, which seems to be a reference to Table 1.

Thank you for pointing that out, indeed we meant to reference the other table at that point and therefore corrected it. 

• Although it can be figured out from the results, it would be helpful to explicitly note in either the “Set of Studies” (p. 10) or the “Data Analysis” section (p. 14) that the correlations are against the sleepiness scale of the RSQ. At the moment, it only says in the “Set of Studies” section that it will be for “each of the three relaxation scales of the RSQ.”

Again, thank you for your feedback. We clarified this as you suggested in the “Data Analysis” section: “Each correlation was against the Sleepiness Scale of the RSQ.” 

(p. 8 line 6).

• The section for “Further analyses” (p. 16) discusses the lack of significant differences but does not discuss the results from the Bayes analyses, despite dedicating two sentences to explaining how to interpret Bayes Factors. A simple sentence on the weak BFs found would be sufficient to help readers who do not look at the Supporting Information.

Thank you for this comment. We followed your advice and added the following sentences to the “Further Analyses” section: “The Bayes Factor mostly supports these findings, indicating moderate evidence only for the significant comparison. Furthermore, two more BF were just above 1, indicating weak evidence for the alternative hypothesis.” (p.10, lines 17 ff.).

• It would be helpful for readers to have titles for the tables and captions for the figures included in the Supporting Information. Apologies if these already exist; they aren’t present in the reviewer version.

We apologize for the inconvenience, however, according to the Journal’s guidelines (“Do not include captions as part of the figure files”), the captions should only be placed in the text.

---

## [Editor Report · Decision Letter 1]

23 Aug 2024

Feeling tired versus feeling relaxed: Two faces of low physiological arousal

PONE-D-24-21576R1

Dear Dr. Steghaus,

We’re pleased to inform you that your manuscript has been judged scientifically suitable for publication and will be formally accepted for publication once it meets all outstanding technical requirements.

**Final editor comments.** The authors have convincingly addressed all points and reworked the manuscript, which is now in excellent shape. I am thoroughly impressed by the attention to detail, as well as the instructive nature of the paper. This dual function to write the manuscript so that it is both a concise tutorial on measurement of state and feelings and an empirical study presenting findings based on data is highly effective. I have reread the manuscript and reviewed the commens of R1, finding everything perfectly addressed. Therefore, after due consideration, I have decided that the manuscript, in its present form, can be accepted.

Kind regards,

Michael B. Steinborn, PhD

Section Editor

PLOS ONE
---

## [Editor Report · Acceptance letter]

29 Aug 2024

PONE-D-24-21576R1 

PLOS ONE

Dear Dr. Steghaus, 

I'm pleased to inform you that your manuscript has been deemed suitable for publication in PLOS ONE. Congratulations! Your manuscript is now being handed over to our production team.

Kind regards, 

on behalf of

Dr. Michael B. Steinborn 

Section Editor

PLOS ONE